# Genetic and Genomic Analysis Identifies *bcltf1* as the Transcription Factor Coding Gene Mutated in Field Isolate Bc116, Deficient in Light Responses, Differentiation and Pathogenicity in *Botrytis cinerea*

**DOI:** 10.3390/ijms26083481

**Published:** 2025-04-08

**Authors:** Virginia Casado-del Castillo, Vlad Paul Mihaila Novac, Alessandro Gabrielli García, José María García Fernández, Paula Iriondo-Ocampo, José María Díaz-Mínguez, Ernesto Pérez Benito

**Affiliations:** Instituto de Investigación en Agrobiotecnología (CIALE), Departamento de Microbiología y Genética, Universidad de Salamanca, C/Río Duero, 12, Villamayor, 37185 Salamanca, Spain; virginiacasado@usal.es (V.C.-d.C.); vpaul@usal.es (V.P.M.N.); alessandrogg@usal.es (A.G.G.); chema1298@usal.es (J.M.G.F.); piriondocampo@usal.es (P.I.-O.); josediaz@usal.es (J.M.D.-M.)

**Keywords:** *Botrytis cinerea* Light Responsive Transcription Factor 1, pathogenicity, light responses, genetic diversity, bulked segregant analysis

## Abstract

Natural populations provide valuable information and resources for addressing the genetic characterization of biological systems. *Botrytis cinerea* is a necrotrophic fungus that exhibits complex responses to light. Physiological analysis of *B. cinerea* populations from vineyards in Castilla y León (Spain) allowed for the identification of isolate Bc116. This field isolate shows a reduced pathogenicity that is conditioned by the light regime. Light also delays germination and accentuates the negative effect it exerts on the vegetative growth of *B. cinerea*. Bc116 also displays a marked hyperconidiation phenotype and a characteristic sclerotia production pattern. Genetic analysis demonstrates that the alternative phenotypes regarding pathogenicity, conidiation, and pattern of sclerotia production co-segregate in the progeny of crosses between isolate Bc116 and a wild-type field isolate, Bc448, showing that they are under the control of a single *locus*. By applying a strategy based on bulked segregant analysis, the mutation in Bc116 has been mapped to a 200 kb region on Chr14 and the analysis of this region identifies a 2 kb deletion affecting the *bcltf1* gene, encoding the *B. cinerea* Light Responsive Transcription Factor 1 described in the reference isolate B05.10. Transformation of Bc116 with the B05.10 *bcltf1* allele restored the wild-type phenotypes, providing functional evidence that the natural mutant Bc116 is altered in gene *bcltf1*. This study offers additional information, derived from the analysis of the genetic background of a natural mutant, on the physiological processes regulated by BcLTF1 and supports the key role of this TF in the pathogenicity and photobiology of *B. cinerea*.

## 1. Introduction

*Botrytis cinerea*, the causal agent of gray mold, is a highly relevant phytosanitary problem in many cultivated plant species. It is a ubiquitous phytopathogenic fungus with a very wide host range, including species of enormous economic importance [1]. It can infect all types of organs and tissues and causes problems both in the field and during post-harvest. Its life cycle is complex. *B. cinerea* may survive in the field as a saprophyte on senescent or decomposing plant tissues or as a pathogen on living plants, causing the death of the host cells to obtain nutrients, a characteristic necrotrophic lifestyle [2,3]. The asexual spores produced on the conidiophores of the mycelium proliferating on the colonized tissues constitute the main structure for dispersion and infection. Mycelium itself is an important source of inoculum in the field. The fungus produces resistance structures, the sclerotia, which allow the pathogen to overcome adverse environmental conditions, such as low winter temperatures. When favorable conditions are restored, the sclerotia germinate, producing a mycelium that actively sporulates. The sclerotia also play a fundamental role in the sexual cycle of the pathogen, acting as a reproductive structure provided by the female parent in crosses in which they are fertilized by microconidia produced by an isolate of compatible mating type that acts as a male parent. The ascospores produced in the derived fruiting bodies, the apothecia, also contribute to the dispersion of the pathogen and, like the spores derived from the asexual phase, are infective [3,4].

The fact that a species presents a sexual phase in its life cycle offers an extremely useful experimental tool in the context of the genetic analysis of traits: the possibility to perform crosses. Early studies on the mating type of *B. cinerea* indicated that it is a heterothallic species [5]. The sexual type is determined by a single *locus* (*MAT1*) with two idiomorphs, *MAT1-1* and *MAT1-2.* Both mating types are widespread in nature [6]. Although infrequently, it is also possible to find pseudo-homothallic isolates that can be crossed with isolates of both sexual types [6,7,8]. Methods to perform crosses under laboratory conditions have been optimized [6,9].

*B. cinerea* is well known as a very plastic organism whose natural populations present very high levels of phenotypic diversity [1]. This variation has been described in relation to diverse physiological aspects, such as vegetative growth, secondary metabolism, resistance to fungicides, virulence, and responses to light [10,11,12,13,14]. Numerous works have also demonstrated, by analyzing molecular variation, that *B. cinerea* populations present a high degree of genetic variability [13,15,16,17]. In recent years, the availability of well annotated reference genomes [18,19] and the resequencing of the genome of numerous field isolates have allowed us to confirm this high genetic variability across the genome [20,21,22]. The phenotypic diversity of the individuals that make up the populations is organized around the pool of genetic variability. Whatever its nature, the existence of variation and the possibility of performing crosses allow for the construction of genetic maps and the development of strategies based on association analysis in segregating offspring to identify genes with a major effect or Quantitative Trait *Loci* (QTLs) involved in determining traits. Bulked segregant analysis (BSA) is a QTL mapping method based on the identification of molecular markers associated with genes involved in the determination of traits of interest [23,24]. It is particularly appropriate in the analysis of offspring populations that show clearly contrasted alternative phenotypes characteristic of the parents involved in a certain cross. Its application has reported successful results in the genetic analysis of non-aggressive mycelial field isolates of *B. cinerea* [25]. On the other hand, the determination of variation in populations of individuals, ideally at the genomic level, makes it possible to carry out Genome-Wide Association Studies (GWASs) that are not dependent on the offspring of crosses. When applied to the *Botrytis*-Arabidopsis pathosystem, indications of a complex architecture of virulence in the fungus were found [26]. A similar trend was observed when investigating the interaction between *B. cinerea* and wild and domesticated tomato genotypes [22]. Candidate polymorphisms and genes associated with virulence have been reported in both systems.

Basic research on the biology of *B. cinerea* and its necrotrophic lifestyle has revealed a close relationship between virulence, development, and light sensing. Early work already indicated that *B. cinerea* can sense light of different wavelengths and that this stimulus regulates differentiation programs. Genome analysis shows that *B. cinerea* possesses 11 photoreceptors whose activities cover the light spectrum (reviewed by Schumacher, 2017) [27]. In nature, most strains respond to light like the B05.10 sequenced reference strain and undergo photomorphogenesis, that is, light exposure stimulates the production of macroconidia while its absence stimulates the production of sclerotia [27]. These strains are classified as light-responsive strains. But natural populations also show variation regarding this ability and “blind strains”, displaying the same phenotype regardless of the light regime applied, “always conidia”, “always sclerotia”, or “always mycelia”, are found. These strains are assumed to be deficient in key components of the light-sensing machinery. Characterization of “always conidia” strains demonstrates that this phenotype may result from alterations in different genes. In the wild strains T4 and 1750, Single Nucleotide Polymorphisms (SNPs) in gene *bcvel1* (member of the VELVET complex) determining stop codons, and therefore generating truncated versions of the encoded protein, were found to be responsible for the observed phenotype [28]. Deletion of any VELVET complex member determined inhibition of sclerotial development, increased conidiation, increased conidial melanogenesis, and, remarkably, reduced virulence [28,29,30]. This demonstrates the role of VELVET in light sensing and development and pathogenicity in *B. cinerea* [29,30]. A similar phenotype has been described in mutants altered in the *B. cinerea* White Collar Complex (WCC), the key component involved in light perception and the coordination of the light response in *B. cinerea*. The functional WCC is the heterodimeric Transcription Factor (TF) integrated by the *bcwcl1* and *bcwcl2* gene products, two GATA-type TFs. The ∆*bcwcl1* mutants generated in the B05.10 background show hyperconidiation and a lack of sclerotial development. They also show precocious and persistent conidiation. Interestingly, they show reduced virulence, but only when incubated in photoperiod conditions, not in permanent darkness. Additionally, the mutants were hypersensitive to oxidative stress caused by H_2_O_2_ [14]. By applying a random mutagenesis approach, another virulence-related factor, BcLTF1 (for *B. cinerea* Light Responsive Transcription Factor 1), was identified and described in the B05.10 strain [31]. The mutants, generated through *A. tumefaciens*-Mediated Transformation (ATMT), were first selected based on their reduced virulence. Detailed physiological and transcriptomic analysis uncovered the functions of BcLTF1 in the regulation of light-dependent differentiation, the equilibrium between production and scavenging of Reactive Oxygen Species (ROS) and secondary metabolism. Specifically, the mutants were unable to produce sclerotia, and were found to hyperconidiate and be hypersensitive to light and oxidative stress. *bcltf1* encodes a GATA-type TF homologous to the *Neurospora crassa* SUB-1 [32] and the *Aspergillus nidulans* NsdD [33]. In these model systems, the TF has been shown to participate in the modulation of light responses and differentiation.

Our group is interested in the characterization of the natural populations of *B. cinerea* from the vineyards of Castilla y León (Spain), the evaluation of their genetic diversity, and the identification of natural mutant strains altered in pathogenicity [13]. As a result, we found several non-aggressive mycelial isolates whose characterization led to the identification of a gene with a major effect on development and pathogenicity in *B. cinerea*, *Bcin04g03490*, initially cataloged as a gene encoding a TF, since the encoded protein has a Gal4-type DNA-binding domain [25]. During these evaluations, a strain with the “always conidia” phenotype was also isolated, which was deficient in its ability to infect *Phaseolus vulgaris* and *Vitis vinifera* leaves. Here, we describe the physiological characterization of this strain, Bc116, and the application of a BSA strategy to map the mutation and identify the altered gene. This analysis demonstrates that strain Bc116 is a natural mutant altered in gene *bcltf1*.

## 2. Results

### 2.1. The Pathogenicity of Bc116 Is Light Dependent

Former studies have indicated that when inoculated plant tissues are incubated under standard photoperiod conditions (16 h light/8 h darkness, LD), isolate Bc116 is completely unable to infect *Vitis* leaves and shows reduced aggressiveness on bean leaves [25]. As light has been shown to affect pathogenicity in *B. cinerea*, we decided to evaluate the effect of the light regime on the behavior of isolate Bc116 in comparison with the aggressive field isolate Bc448 [25]. Pathogenicity assays were carried out on *V. vinifera* leaves, variety Juan García, by inoculating mycelium plugs from colonies actively growing on MEA plates. When the inoculated leaves entered the photoperiod at the beginning of the light phase (LD), isolate Bc116 was shown to be unable to cause infection (Figure 1A-LD), as shown previously. The same behavior was observed when incubated under permanent light conditions (LL) (Figure 1A-LL). When the leaves were incubated under permanent darkness conditions (DD), isolate Bc116 was as infectious as the aggressive isolate Bc448 (Figure 1A-DD,B). Interestingly, when the inoculated leaves entered the photoperiod at the beginning of the dark phase (DL), isolate Bc116 was also able to cause infection (Figure 1A-DL). These results indicate that light blocks the ability for isolate Bc116 to infect the *Vitis* leaves, but that it is specifically the effect of light during the early stages of the plant–pathogen interaction that conditions the ability of the fungus to cause infection.

The effect of light was also evaluated on bean leaves. In this case, inoculations with isolates Bc116 and Bc448 were performed on whole plants. At 24 h post-inoculation (hpi) (Figure 2A), the isolate Bc116 did not produce any necrotic lesion on the leaves when the plants were incubated under LL regime. When incubating the inoculated plants under LD conditions, a very limited infection capacity of isolate Bc116 was observed in comparison with the aggressive Bc448 isolate. At this time point, the necrotic lesions produced by Bc116 were slightly smaller than those caused by Bc448 under the DL and DD conditions. The microscopic observation of stained tissues correlated with these observations (Figure 2B). Monitoring the progress of infection with time showed that Bc116 can establish an interaction with the host tissues, but that infection is severely delayed in comparison with isolate Bc448: at 72 hpi, expanding necrotic lesions were generated by Bc116 in all light regimes, but the diameter of the lesions was smaller under LL and LD conditions than under DL and DD conditions (Figure 2C,D). Bc448 generated larger necrotic lesions than Bc116 under the four light regimes (Figure 2C,D). When incubation was extended, both isolates produced full maceration of the plant tissues.

### 2.2. Light Limits Bc116 Ability to Penetrate Onion Epidermal Cells

As Bc116 shows a delay in infection in the presence of the light stimulus, we decided to evaluate possible alterations in the processes involved in early stages of infection. Specifically, we decided to evaluate its ability to penetrate onion epidermis cells in inoculations performed either with a spore suspension or with mycelium plugs. At 12 hpi, spores of the aggressive Bc448 isolate were found to have germinated and penetrated the cells, and they essentially did to the same extent under both LL and DD (Figure 3E,G). The Bc116 spores germinated and penetrated the cells under DD conditions (Figure 3C), but under LL conditions, although germination of the spores was observed, penetration was hardly detected. Instead, non-germinated spores and short and long germ tubes were observed on the cells’ surface (Figure 3A). When inoculations were performed with mycelium plugs, Bc448 produced branching mycelia inside the epidermal cells at 24 hpi, both under LL and DD conditions (Figure 3F,H). However, Bc116 showed penetrating or expanding mycelium inside the cells only under DD conditions (Figure 3D). In longer incubations (48 and 72 hpi), both with conidia and mycelium, Bc116 showed hyphae inside the host cells in LL. Taken together, these observations indicate that light conditions the ability of isolate Bc116 to penetrate the host cells and determines a delay in this process.

### 2.3. Light Affects Germination, Vegetative Growth Rate, and Conidiation in Bc116

The infection process of *B. cinerea* requires the germination of conidia and subsequent penetration in the plant tissues. If Bc116 shows a limited ability to infect when the initial stages of the infection process occur under light exposure, could the germination program of Bc116 be affected by light? To evaluate this, a germination assay in liquid synthetic medium was performed. Drops of conidial suspensions were placed in the center of empty Petri dishes and samples were incubated at 22 °C under LL or DD conditions. Estimations of germination were carried out after 6 h, by which time most of the spores have germinated and lengths of the germ tubes can easily be scored. As shown in Figure 4A, the progress of germination of the Bc448 isolate was very similar under both light conditions, since the percentages of spores in each of the three considered stages was very similar under LL and DD. In Bc116, the light exposure determined a delay in the germination process, as the percentage of conidia in stage 2 was significantly higher when the fungus was incubated under DD than under LL. The situation was the opposite for stages 0 and 1.

Bc116 has been reported to be a hyperconidiating isolate, but its capacity to sporulate has not been quantified and it has not been determined whether light influences sporulation. To investigate this, we determined the production of spores by isolate Bc116 in MEA plates in comparison with the field isolate Bc448 under the three light regimes, LL, LD, and DD. It was decided to include the B05.10 strain in this analysis to gain information in comparison with a reference isolate. In B05.10, light was shown to stimulate the production of spores, resulting in higher sporulation under continuous light. Bc116 was found, indeed, to produce very large amounts of spores, between one and two orders of magnitude higher than those produced by isolates Bc448 and B05.10, and it hyperconidiated in the three light regimes, although more efficiently under DD conditions (Figure 4B). This implies that in Bc116, the light stimulation of sporulation was not manifested. Remarkably, the Bc448 field isolate resembled the reference strain B05.10 regarding the number of spores produced, but the stimulating effect of light was not manifested either. In this regard, the two field isolates Bc116 and Bc448 displayed a similar response.

The effect of light on the saprophytic growth rate was also investigated. To this end, fungal cultures on MEA plates were initiated with a drop of a spore suspension. After 5 days of incubation at 22 °C under the three light regimes, the colony diameter was estimated. The growth rate was negatively affected by light in B05.10 and Bc448, as shown by the reduction in the colony diameter (reduction of 4.46% and 4.83% for isolate Bc448 under LL and LD, respectively; reduction of 4.41% and 5.48% for B05.10 under LL and LD, respectively) (Figure 4C). Bc116 grew more slowly than the B05.10 and Bc448 isolates in the three light regimes, but light further slowed down the growth of Bc116, indicating that it negatively affects the growth of this strain. The effect of light on Bc116 under LL or LD was quantitatively similar, as the growth reduction was 12.15% and 13.12%, respectively. Although the saprophytic growth rate was reduced in all light regimes, Bc116 showed earlier sporulation both in cultures initiated either with spores or mycelium plugs (see Figure 5), as already evident in four days old cultures.

The production of sclerotia in most *B. cinerea* isolates was stimulated in the absence of light and was favored by low temperatures. In these conditions, Bc448 and B05.10 isolates produced large, irregularly distributed sclerotia on the plates. Bc116 also produced sclerotia, but following a different pattern characterized by the appearance of large numbers of very tiny sclerotia, most deeply embedded in medium and distributed all over the plate (Figure 4D).

### 2.4. Bc116 Is Hypersensitive to Oxidative Stress Under LL Conditions

As *B. cinerea* isolates altered in light responses are often reported to be also altered in their capacity to cope with oxidative stress [14,31], we evaluated their sensitivity to the oxidative stress conditions derived from the exposition to H_2_O_2_ under the three light regimes considered. To this end, cultures were initiated by transferring mycelium plugs to the center of MEA plates containing 7.5 mM H_2_O_2_ and then incubated at 22 °C under LL, LD, and DD. Sensitivity to H_2_O_2_ was quantified by measuring the colony diameter 96 hpi. The three strains displayed reduced growth in the presence of H_2_O_2_ under the three light regimes. The behaviors of B05.10 and Bc448 were very similar and the magnitude of the reduction they manifested under the three light regimes was nearly the same (31.4%, 31.3% and 34.6% for B05.10 under LL, LD and DD, respectively; and 29.9%, 29.2% and 25.7% for Bc448 under LL, LD and DD, respectively). Bc116 was found to be more sensitive to oxidative stress, particularly under LL conditions (growth reduction of 69.7%, 63.1%, and 40.5% under LL, LD, and DD, respectively) (Figure 5).

### 2.5. Genetic Analysis of the Bc116 Phenotype

Bc116 displays alterations in several aspects related to pathogenicity, development, and differentiation and responses to the light stimulus. To characterize the genetic basis of these alterations, we undertook a genetic analysis. The Bc116 isolate has been found to carry the *MAT1-2* allele and the Bc448 isolate—a highly aggressive field isolate belonging to the same natural population (vineyards of Castilla y León, Spain) [13], which is being used in our studies as a reference isolate and resembles the reference isolate B05.10 in most aspects—has been shown to carry the *MAT1-1* allele [25]. Therefore, Bc116 and Bc448 show contrasting phenotypes and should be sexually compatible. Crosses between the two isolates were attempted. As Bc116 produces tiny sclerotia, crosses could only be established using Bc116 as the spermatizing strain, as it produces microconidia. An offspring was collected from the ♂Bc116 × ♀Bc448 cross, consisting of 222 single ascospore isolates. First, the traits “aggressiveness” and “pattern of conidiation” were scored in the full set of descendants 122 individuals were shown to hyperconidiate and be unable to infect *Vitis* leaves, while 100 displayed normal sporulation patterns and caused infection. When incubated at low temperatures in DD, the non-aggressive and hyperconidiating isolates were found to produce sclerotia like the Bc116 isolate, while the aggressive isolates which sporulate normally all produced sclerotia like the Bc448 isolates (Appendix A shows the phenotypes of a representative selection of the progeny). These observations indicate that the three traits co-segregate, and the proportions observed informed of a 1:1 segregation. Therefore, it can be concluded that the three traits considered, “aggressiveness”, “pattern of conidiation”, and “pattern of sclerotia production”, are under the control of a single genetic *locus*, which has been altered in the natural isolate Bc116. It is interesting to note that large differences in aggressiveness were observed among the individuals integrating the aggressive progeny (Appendix A). This observation indicates that the altered gene in the natural isolate Bc116 regulates, in addition to genetic factors involved in light responses and differentiation, several pathogenicity-related genes present in different configurations in the Bc116 and Bc448 backgrounds, which segregate in the aggressive progeny.

### 2.6. Mapping the Altered Gene in Bc116 by BSA

A BSA strategy was applied to the segregating population from the ♂Bc116 × ♀Bc448 cross in order to identify molecular polymorphisms co-segregating with the alternative phenotypes of the traits of interest. We previously obtained information about the sequence polymorphisms of isolate Bc448 in the form of SNPs in comparison with the genome sequence of the reference isolate B05.10 [25]. We sequenced the Bc116 genome, and high-quality short reads were aligned to the B05.10 genome reference sequence to identify the SNPs in Bc116. Analysis of the list of the polymorphisms of each isolate identified 83,118 SNPs exclusive to isolate Bc448 and 114,893 SNPs exclusive to isolate Bc116. Table 1 shows the chromosomes described in the B05.10 genome and their sizes, as well as the total number of SNPs identified in each chromosome in the genomes of Bc448 and Bc116 and the number of SNPs exclusive to one isolate or the other. The minichromosome Chr18 described in B05.10 [18] was not present either in the Bc448 or in the Bc116 genomes.

Two groups of descendants, A and B, from the cross ♂Bc116 × ♀Bc448, were established, each one integrated by 60 individuals: the first one included individuals resembling the aggressive isolate Bc448 and the second one individuals resembling the non-aggressive isolate Bc116. Then, the distribution of the frequencies of the SNPs specific of each parental isolate was analyzed in the two groups of individuals. Except in Chr14, the plots of the SNP index display values close to “0” along the entire chromosome for most chromosomes, as shown for Chr13: (Figure 6A, plots for other chromosomes show the same profile). In Chr14 the two SNP plots peaked around coordinate 1,500,000, reaching maximal values (close to +1) for the one corresponding to the Bc116 isolate specific SNPs and minimal values (close to −1) for the one corresponding to the Bc448 isolate specific SNPs. Both plots delimitated a genomic region of markers associated with the segregating phenotypes of about 200 kb (between coordinates 1,400,000 and 1,600,000) (Figure 6B). This region includes 41 annotated genes in the genome of the reference strain B05.10 (Appendix A).

The mapped region was analyzed in detail, searching for polymorphisms with an expectedly large impact that could determine a loss of function and explain the observed phenotypes. As a second criterion, genes with a predicted regulatory role were considered as best candidates, as the mutation in Bc116 determines alterations in diverse processes. A list of the genes annotated in the mapped region, together with the polymorphisms identified, is presented in Appendix A.

Among the candidates, the *bcltf1* gene (*Bcin14g03940)* was detected, described previously as a gene encoding a TF regulating virulence and light responses in *B. cinerea* [31]. In the Bc116 genome, a large deletion of about 2 kb in size was identified (Figure 6C) between positions 1,530,248 and 1,532,273. This deletion involves the 5′-UTR of gene *bcltf1* and the first 211 nucleotides of its structural region, including the first exon, consisting of 32 nucleotides, the first intron, consisting of 91 nucleotides, and the first 88 nucleotides of the second exon. Because of this mutation, the encoded protein is expected to lack the first 54 amino acids. The presence of the deletion and its extension were confirmed by amplifying the deleted region with primers flanking it in the reference genome (Figure 7A) and sequencing it.

### 2.7. Bc116 Is Altered in the Light Responsive Transcription Factor BcLTF1

Given the functions reported for *bcltf1* and the nature of the mutation identified, it was selected as the first candidate to be considered as the gene altered in Bc116. To determine if the mutation identified in *bcltf1* is responsible for the phenotype observed in isolate Bc116, a strategy based on functional complementation was undertaken. To this end, a 4.7 kb genomic DNA fragment containing the entire wild-type allele of *bcltf1*, including the sequence deleted in the genome of Bc116, was amplified by PCR with primers *bcltf1-c2F* and *bcltf1-c2R* (annealing positions are indicated in Figure 6C) using as the template genomic DNA from the reference isolate B05.10. The fragment was cloned into plasmid pWAM6, giving rise to plasmid pVPM1, and from it a 7.7 kb linear fragment containing the *bcltf1* wild-type allele and the hygromycin resistance cassette was mplified with primers *bltf1-c2R* and *PoliC-F5′*. The fragment was transformed into Bc116 protoplasts. Two independent transformants, T2 and T7, were selected, in which the *bcltf1* wild-type allele, together with the mutant allele, could be detected (Figure 7A). These two transformants were found to be able to infect *Vitis* leaves under LD conditions, behaving like the Bc448 and B05.10 isolates (Figure 7B), displayed growth rates similar to those of the wild type strains, did not show premature sporulation in the MEA plates (Figure 5 and Figure 7C), and did not hyperconidiate. Neither of the transformants showed increased sensitivity to the oxidative stress derived from the addition of H_2_O_2_ (Figure 7C) (growth reduction of 32.1%, 34.9%, and 29.8% for T2 under LL, LD, and DD, respectively; and 29.2%, 31.3%, and 32.6% for T7 under LL, LD, and DD, respectively; these are values similar to those observed for the B05.10 and Bc448 isolates and smaller than those observed for the Bc116 isolate) and they resembled B05.10 and Bc448 more than Bc116 in their pattern of sclerotia production (Figure 7D). All these observations show the restoration of the wild-type phenotypes in T2 and T7 transformants, thus demonstrating the functional complementation.

## 3. Discussion

### 3.1. Genetic and Genomic Tools Facilitate Association Mapping in B. cinerea and Identify a Natural Variant of bcltf1

*B. cinerea* is considered one the most important fungal pathogens [34]. Experimental evidence is accumulating, outlining the mode of action of a pathogen that establishes interaction with the host by manipulating and exploiting essential biological processes of the plant for its own benefit [35]. Recent research shows that *B. cinerea* has a remarkable capacity to sense light stimuli, which the fungus integrates to make developmental decisions, and several authors consider *B. cinerea* to be a valuable model system to expand our knowledge of fungal photobiology [14,27]. Our work supports the existence of close relationships at the level of genetic determination and regulation between the processes involved in pathogenicity and responses to light, since the identified natural mutants showed an altered ability to infect the host plant, and Bc116, among them but not alone [25], also showed alterations in its responses to light.

Genetic analysis demonstrates that although Bc116 shows alterations in different aspects related to pathogenicity, development, and responses to light, these impacts all depend on a single gene. This situation facilitates the consideration of procedures such as BSA to map the mutation and identify the altered gene. Although originally developed to map and identify QTLs [24], BSA is particularly well suited for the identification of genes with a major effect on the phenotype(s) of interest. Our group previously successfully applied this methodology to *B. cinerea* [25]. In the work here presented, Bc116 is shown to harbor levels of polymorphism like those reported for the Bc448 isolate [25] and for other field isolates [20,22]. By adapting the experimental framework previously considered, it was possible to perform an association mapping covering the entire genome of our *B. cinerea* isolates, offering a resolution that enables mapping the mutation in isolate Bc116 to a 200 kb genomic region in Chr14. The availability of a fully sequenced and accurately annotated reference genome has become an invaluable tool. The detailed analysis of the mapped region considering the two criteria highlighted, a mutation expected to have high impact and affecting a genetic factor with possible regulatory function, led to the selection of *bcltf1* as the most likely candidate. The restoration of the wild-type phenotypes by transformation with the allele derived from strain B05.10 demonstrates that *bcltf1* is the altered gene in Bc116.

### 3.2. Bc116 Resembles B05.10-Δbcltf1

Former studies performed by Schumacher et al. [31] identified and described BcLTF1 as a virulence factor in *B. cinerea*. They generated mutants in B05.10 that displayed reduced aggressiveness on bean plants and alterations in several differentiation and development processes regulated by the light stimulus. Furthermore, they were found to be hypersensitive to exogenously applied oxidative stress. Isolate Bc116, like the B05.10-Δ*bcltf1* mutants, showed a reduction in growth in all light regimes, and this reduction was greater under permanent light conditions. It also shares the early sporulation and hyperconidiation phenotypes with them, but its capacity to produce spores is even greater. It is interesting to note that Bc116 does not show a stimulation of sporulation by light, described as characteristic of *B. cinerea* isolates [27]. This is also observed in the field isolate Bc448, supporting the existence of a natural variation range related to these responses.

### 3.3. Characterization of Bc116 Broadens the Knowledge of the BcLTF1 Functions

A remarkable difference between Bc116 and the B05.10-Δ*bcltf1* mutants is the ability of Bc116 to produce sclerotia. Under DD and low temperatures, for about one month, Bc116 can produce sclerotia, although the production pattern is different to that displayed by Bc448 and B05.10. These conditions are similar to those reported in the characterization of the B05.10-∆*bcltf1* mutants [31]. Therefore, it is likely that the differences observed between the B05.10-∆*bcltf1* mutants and Bc116 are due to their different genetic backgrounds. The pattern of sclerotia formation is controlled by BcLTF1, as shown by the segregation observed in the offspring of the Bc116 × Bc448 cross and the pattern exhibited by complemented transformants, which resembles that of Bc448. On the other hand, although the size of the sclerotia of Bc116 limits the possibility of their utilization in crosses, this isolate was successfully used as the male parental strain in our crosses, demonstrating that it is sexually competent.

The behavior of Bc116 in the inoculations on bean leaves in LD was similar to that described for the B05.10-∆*bcltf1* mutants under the same conditions: the progress of infection was delayed. However, when extending the analysis to other light regimes, it is found that the factor limiting the mutant’s capacity to cause infection is the light exposure during the early phases of the plant–fungus interaction. In the B05.10-∆*bcltf1* mutants, no differences were observed in their ability to penetrate onion epidermis in DD conditions in comparison with the wild type, and it was concluded that the delay in infection was due to limitations in colonization, not in penetration. Our analysis detects the same situation in DD, but it is verified that in LL, light reduces the efficiency of penetration into the onion epidermis of both spores and mycelium. This indicates that the Bc116 isolate has a limited capacity to cope with the effect of light during the early stages of development, which in planta involves penetration. The work by Schumacher et al. [31] found that their B05.10-∆*bcltf1* mutants are hypersensitive to oxidative stress. The authors cleverly demonstrated that the delay in the infection process and the reduced growth observed in planta was due to the limited capacity of the *bcltf1* mutants to cope with the oxidative stress that arises during light exposure. Bc116 is also hypersensitive to the oxidative stress produced by exposure to H_2_O_2_ during saprophytic growth. This sensitivity is increased in LL and LD, which confirms the above-mentioned observations.

The infection capability of Bc116 is more limited on *Vitis* leaves than on *P. vulgaris* leaves. Since the sensitivity to oxidative stress accounts for the reduction in aggressiveness on the common bean, it could be inferred that the production of ROS in *Vitis* is higher. Alternatively, *Vitis* may produce metabolites that added to the oxidative stress conditions created during the early stages of the interaction, limiting the ability of the natural *bclft1* mutant to cause infection. It is worth noting that B05.10-∆*bcltf1* mutants show a severely altered secondary metabolism [31] that may interact with the defense metabolites produced in *Vitis* and *P. vulgaris* in different ways.

It is striking that on *Vitis* leaves, the effects of light exposure during the early stages of the interaction determined alterations in the fungus that were maintained over time once the inoculated tissues entered the darkness conditions (the dark phase in LD). This implies that during the first moments of the attempted infection, the conditions created by the presence of light in the environment in which the plant tissues interact with the infective structures of the fungus determine irreversible alterations in the development program of the pathogen. It is interesting to note that ROS play crucial roles in development and differentiation processes of fungi [36]. The interaction described in this work may offer an interesting experimental system to identify specific targets of the ROS that are key in the definition of these differentiation programs in *B. cinerea*.

### 3.4. The bcltf1 Allele in the Field

Field populations constitute a source of natural variation, both in pathogenicity [25] and in light responses [14]. The presence of isolates with reduced or null pathogenicity on plant tissues raises questions concerning their ability to compete with fully virulent isolates. A hyperconidiating phenotype, as the one displayed by the *bcltf1* mutants, might be expected to be advantageous, as it strongly favors the dispersal of the pathogen. However, the severe limitation in counteracting the toxic effects of light, a stimulus that can hardly be avoided in the field, which delays germination and penetration and reduces vegetative growth, together with their hypersensitivity to ROS, is expected to negatively affect their fitness. This balance would determine the frequency of the mutant allele in the population.

To our knowledge, Bc116 is the first natural mutant altered in *bcltf1* to be reported in the literature. Our group has been monitoring the incidence of gray mold in the vineyards of Castilla y León (Spain) since 2000. We have evaluated a collection of 1050 isolates to analyze variations in pathogenicity. As a result, only one *bcltf1* natural mutant has been found, while seven mycelial non-pathogenic mutants harboring different loss of function mutations in the gene *Bcin04g03490*, the ortholog of the *N. crassa ff-7* gene, have been isolated [25]. These observations indicate a low frequency of the *bcltf1* mutant allele in the field, suggesting that alteration in BcLTF1 functions reduces the fitness of the natural mutants. However, their survival ability in vineyards points towards some ecological niche where they successfully compete with fully virulent isolates.

## 4. Materials and Methods

### 4.1. Organisms and Growth Conditions

The *B. cinerea* isolates B05.10 [37], Bc116, and Bc448 [13] were used in this study. Fungal cultures were established from frozen conidia stored on 15% glycerol (*v*/*v*) at −80 °C. Fungal isolates were grown at 22 °C under the light/darkness conditions required for each experiment. Light was generated by Cool White Osram L 36 W/840 fluorescent bulbs.

Common bean plants (*Phaseolus vulgaris* L.) cv Blanca Riñón, a local landrace, were kindly provided by Centro de la Legumbre (Pajares de la Laguna, Salamanca, Spain). Plants were grown in natural substrate for 2 weeks in the greenhouse at 20–24 °C under a 16/8 h light/darkness photoperiod. The *Vitis vinifera* plants variety Juan García were maintained in the greenhouse under the same conditions.

### 4.2. Germination, Conidiation, and Saprophytic Growth Experiments

To study the fungal germination patterns, the strains were grown on MEA (ThermoFisher Scientific Inc., Waltham, MA, USA) plates for 3 weeks at 22 °C in permanent darkness. Conidia were harvested and suspensions of 5 × 10^5^ conidia/mL were prepared in PDB (Potato Dextrose Broth, Difco) at half concentration. Further, 60 µL drops of the conidial suspensions were placed in the center of empty Petri dishes that were incubated without agitation inside a wet chamber at 22 °C under continuous light or continuous darkness. Samples were imaged at 6 hpi using a MD-E3-6.3 camera (MicrosCopiaDigital, Industrial Digital Camera) adapted to a microscope Leica DLMB (Leica Microsystems, Bensheim, Germany). The percentage of germinated conidia was quantified according to a previously described classification of the conidia developmental stages [38]. Five plates per condition and strain were analyzed in each experiment and experiments were repeated three times.

To analyze conidiation rates, the fungal isolates were grown on MEA for 4 days at 22 °C under permanent darkness. Agar plugs 5 mm in diameter from the edge of the colony were taken and placed in the center of Petri dishes containing MEA. The plates were maintained at 22 °C and under different light conditions (LL, 16/8 h LD, DD) for 3 weeks, and then the conidia were harvested from each plate. The number of conidia produced was estimated using a Thoma cell counting chamber. Three plates per strain and experiment were analyzed and three independent biological experiments were carried out.

Fungal saprophytic growth was determined on MEA. Fungal isolates were grown, and conidia suspensions were prepared as described above for the germination assays; 10 µL drops of the conidial suspension were placed in the center of Petri dishes containing MEA. The plates were incubated at 22 °C for 5 days and then the diameter of the colonies was measured. Different light conditions were assayed: LL, LD and DD. Three plates per strain and light condition were inoculated in each biological experiment and three independent experiments were performed. For the evaluation of the effect of oxidative stress, the isolates were grown on MEA supplemented with 7.5 mM of H_2_O_2_ using agar mycelium plugs 5 mm in diameter as the initial inoculum and were measured the colony diameter 96 hpi. Three plates per strain and condition (medium with and without H_2_O_2_, and light condition, LL, LD, or DD) were inoculated in each biological experiment and three independent experiments were carried out.

The capacity of the fungal isolates to produce sclerotia was evaluated on MEA plates. Mycelium plugs taken from the edge of actively growing colonies were placed in the center of Petri dishes containing the media and incubated at 22 °C for 4–5 days under permanent darkness. Afterwards, the plates were incubated at 2–4 °C under DD conditions for 4 weeks, and then the number, size, and distribution of sclerotia were analyzed. Three plates per strain were inoculated in each biological experiment, and three independent experiments were analyzed.

### 4.3. Penetration Analysis

The ability of fungal isolates to penetrate host tissues was analyzed on onion epidermal cells. Strips of onion epidermis were cut and placed on a slide with the hydrophobic layer side on top. Then, 10 µL drops of a conidial suspension at 5 × 10^4^ conidia/mL prepared in water, or non-sporulating mycelium plugs of 3 mm in diameter taken from the edge of actively growing colonies, were placed on the strips. The samples were maintained at 22 °C under the light conditions required (LL and DD) inside closed plastic boxes to ensure a high-humidity environment. At the time of analysis, the mycelium plugs were removed. In both inoculations, 10 µL of lactophenol blue solution (Fluka, SIGMA-Aldrich, Darmstadt, Germany) was placed on each inoculation spot and incubated for 15 min at room temperature. Afterwards, the staining solution was removed, and the samples were washed with distilled water. Samples were imaged using a MD-E3-6.3 camera (MicrosCopiaDigital, Industrial Digital Camera) adapted to a microscope Leica DLMB (Leica Microsystems, Bensheim, Germany). For each type of inoculum, three strips of onion epidermis and five inoculation spots per strip were used per strain and light condition in each biological experiment, and experiments were repeated three times.

### 4.4. Inoculation Assays

The pathogenicity of fungal isolates was studied according to previously described methods [13,25]. The inoculation on common beans was performed with whole plants that were placed inside transparent plastic boxes with water on the bottom; the inoculation on *Vitis* was carried out on detached leaves whose petioles were inserted in wet floral foam and then placed inside plastic trays with wet paper on the bottom. In both systems, plugs of fresh mycelium 5 mm in diameter taken from the edge of fungal colonies actively growing on MEA plates were placed on non-wounded leaves. Four plugs and one isolate were used per leaf in the case of common bean inoculations, and four plugs and two isolates were used per leaf in the case of *Vitis* leaves. At least 5 leaves were inoculated per fungal isolate and condition in each experiment. The experiments were repeated in a randomized design (i.e., different sets of common bean plants and leaves from different *Vitis* plants were used, and the position of the plastic boxes and trays in the growth chamber was changed in each experiment) at least three times. The inoculated materials were maintained in closed boxes or trays at 22 °C under the light conditions required in each case. Aggressiveness was evaluated by measuring the diameter of the lesions 3 and 4 dpi for common bean plants and *Vitis* leaves, respectively.

For the staining of inoculated common bean leaves, the mycelium plugs were removed 24 hpi and the area of the inoculation was cut. Samples were immersed in a solution of lactophenol blue–ethanol (1:2) and incubated for 1 min at 100 °C, cooled at room temperature, and incubated again for 30 s at the same temperature. After cooling at room temperature, the staining solution was removed, and the samples were washed with absolute ethanol. Images were acquired using a Leica DFC495 camera adapted to a Leica 205FA stereomicroscope (Leica Microsystems, Bensheim, Germany) and analyzed using the LAS software v3.6.0 (Leica Microsystems, Bensheim, Germany). Eight inoculation spots per strain and light condition were cut, stained, and analyzed in each biological experiment, and experiments were performed three times.

### 4.5. Standard Molecular Techniques

Fungal genomic DNA was isolated from mycelium cultured on cellophane sheets placed onto MEA plates. All the DNA purifications followed previously described procedures [39].

The PCR reactions were carried out using the DNA Polymerase from Biotools B&M Labs (Madrid, Spain), except for the amplification of the *bcltf1* allele which was performed using the Phusion High Fidelity DNA polymerase from ThermoFisher Scientific Inc. (Waltham, MA, USA). In both cases, the reactions were performed according to the manufacturer’s recommendations. All the primers used in this work are listed in Appendix A.

The Gateway BP Clonase II Enzyme Mix (ThermoFisher Scientific Inc., Waltham, MA, USA) was used in the cloning reactions of the *bcltf1* allele in plasmid pWAM6 [25].

### 4.6. B. cinerea Transformation

*B. cinerea* protoplasts were transformed using the method described by ten Have et al. [40], with modifications as specified by Reis et al. [41] and Leisen et al. [42]. Protoplasts were generated using a 1% concentration of Vinotaste Pro (Lamothe Abiet, Canejan, France), an enzymatic blend of chitinases and glucanases, along with 0.1% Yatalase (Takara, Saint-Germain-en-Laye, France).

### 4.7. Construction of Bc116-bcltf1-Complemented Transformants

The wild-type allele of *bcltf1* from the B05.10 strain was cloned into plasmid pWAM6 [25], which contains a hygromycin (*Hph*) resistance cassette from pOHT [43], using the Gateway cloning technology. To this end, the B05.10-*bcltf1* allele was amplified as a 4.7 kb fragment with specific oligonucleotides harboring extensions with *attB* sequences (Appendix A). Upon recombination facilitated by the BP clonase, the plasmid pVPM1 was generated. From it, a 7.7 kb linear fragment containing the B05.10-*bcltf1* allele and the *Hph* cassette was amplified using specific oligonucleotides (Appendix A). The PCR product was used to transform Bc116, and transformants able to grow on selective media were successively transferred to fresh selective plates. Monosporic cultures were obtained from the selected transformants.

### 4.8. Crosses

Crosses between Bc448 and Bc116 were performed following the procedures established by Faretra et al. [6,9]. Mature apothecia were collected and crushed in water to release the ascospores. The suspension was filtered through glass wool and plated on MEA plates. Individual ascospore germlings were transferred 24 h later to fresh MEA plates for propagation.

### 4.9. Sequencing and Determination of Polymorphisms

For this study, four data sets were used. The genome of Bc448 has been previously sequenced [25] (Accession number SRR13700579). The genomic DNA of the parental strain Bc116 and of both descendant groups—those resembling Bc448 and those resembling Bc116—in the cross Bc116 x Bc448 was sequenced using Illumina technology by Novogene (Cambridge, UK). The trimming and the quality control were performed with FASTP. The clean sequences were mapped to the genome of the reference isolate, B05.10 (ASM14353v4), using the mapper Bowtie2, and polymorphisms were subsequently extracted using a specific tool for SNP calling from Geneious Prime^®^ 2023.1.1 (Biomatters, Auckland, New Zealand). Further analyses of the tables of polymorphisms were performed with Microsoft Excel.

### 4.10. BSA

For the BSA, a list of SNPs of each parental isolate in comparison with the B05.10 reference genome was generated with Geneious Prime^®^ 2023.1.1 (Biomatters, Auckland, New Zealand). From these, two lists of SNPs exclusive to either Bc448 or Bc116 were derived. From the progeny, two groups of individuals were selected: Group A, consisting of 60 individuals resembling the parental isolate Bc448, and Group B, consisting of 60 individuals resembling the parental isolate Bc116. Genomic DNA was extracted from each individual isolate, and equal amounts of DNA from the 60 individuals in each group were pooled together to form two bulk DNA samples. These genomic DNA pools were sequenced by Illumina, and the frequencies of the polymorphisms specific to either Bc448 or Bc116 were determined in each pool. For the association mapping analysis, only high-quality SNPs (quality score > 33) were considered. The distribution of polymorphisms specific to each parental isolate in the two progeny groups was analyzed by calculating the difference in the frequency of each polymorphism between DNA pool B and DNA pool A (f “polymorphism x” in B—f “polymorphism x” in A). This difference generated an SNP index (*Y*-axis) that was plotted against the chromosomal coordinates (*X*-axis) of the reference genome B05.10. For markers unlinked to the *locus* responsible for the phenotypic difference, allelic frequencies were expected to be similar in both pools, causing the SNP index plot for these chromosomal regions to fluctuate around the “0” value. Conversely, for markers linked to the *locus* of interest, allelic frequencies differed significantly between the DNA pools, with larger differences observed for markers more closely linked to the *locus*. The SNP index reached maximum values (close to +1) for Bc116-specific alleles predominantly found in the non-aggressive, hyperconidiating progeny DNA pool and minimum values (close to −1) for Bc448-specific alleles predominantly found in the aggressive progeny DNA pool.

### 4.11. Statistical Analysis

All the statistical analysis were performed with the help of the software Statistix 10 (Analytical Software, Tallahassee, FL, USA).

## 5. Conclusions

The isolation of Bc116, a natural mutant of gene *bcltf1*, from a collection of isolates recovered from vineyards of Castilla y León (Spain) confirms the power of the battery of genetic and genomic tools established and available in *B. cinerea* to identify the genetic factors behind natural variation in pathogenicity.

The natural mutant resembles the formerly analyzed B05.10-∆*bcltf1* mutants. In the two genetic backgrounds, the mutation causes alterations in light dependent developmental processes and pathogenicity, together with hypersensitivity to oxidative stress, demonstrating a central role for BcLTF1 in the regulation of these processes. Our work provides additional information on how light limits the capacity of the fungus to cause infection and shows additional functions regulated by BcLTF1, thus enhancing its critical role in the regulation of light responses, differentiation, and pathogenicity in *B. cinerea*.

## Figures and Tables

**Figure 1 ijms-26-03481-f001:**
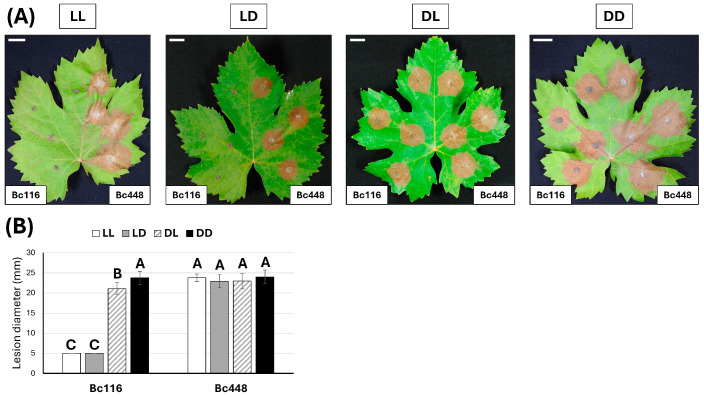
Evaluation of the effect of light on the pathogenicity of isolates Bc116 and Bc448 on *Vitis* leaves. (**A**) *Vitis* leaves were inoculated with mycelium plugs of both strains and incubated under the indicated light regimes for 96 h. The 16/8 h photoperiod regime was initiated either in the light phase (LD) or in the darkness phase (DL). Scale bars, 20 mm. (**B**) Quantification of aggressiveness of isolates Bc116 and Bc448 on *Vitis* leaves under different light conditions (permanent light -LL-, white bars; 16/8 h light/darkness photoperiod -LD-, gray bars; 8/16 h darkness/light photoperiod -DL-, striped bars; permanent darkness -DD-, black bars) estimated as the medium lesion diameter. Bars show the media ± standard deviation (SD) of three independent biological experiments. The letters over each bar represent significant differences between all the conditions assayed that were tested using an ANOVA analysis followed by Tukey’s HSD test (*p* < 0.05).

**Figure 2 ijms-26-03481-f002:**
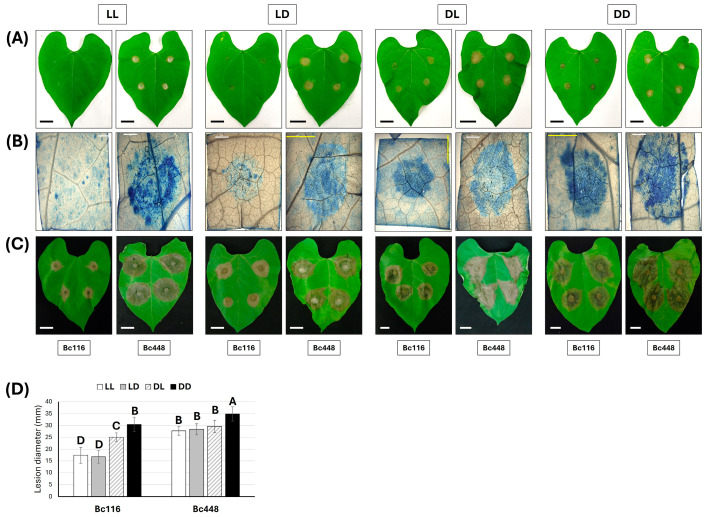
Evaluation of the effect of light on the pathogenicity of isolates Bc116 and Bc448 on bean leaves. Bean leaves were inoculated with mycelium plugs of both strains and incubated under the indicated light regimes. (**A**) Aspect of the inoculated leaves 24 hpi. Scale bars, 20 mm. (**B**) Lesions generated by Bc116 and Bc448 24 hpi. At the indicated time point, the mycelium plugs were removed, and the plant tissues were stained with lactophenol blue. Images were taken using a Leica DFC495 camera adapted to a Leica 205FA stereomicroscope (Leica Microsystems, Bensheim, Germany). White scale bars, 2 mm; yellow scale bars, 5 mm. (**C**) Aspect of the inoculated leaves 72 hpi. Scale bars, 20 mm. (**D**) Quantification of aggressiveness of isolates Bc116 and Bc448 on bean leaves under different light conditions (LL, white bars; 16/8 h LD, gray bars; 8/16 h DL, striped bars; DD, black bars) estimated as the medium lesion diameter 72 hpi. Bars show the media ± SD of three independent biological experiments. The letters over each bar represent significant differences between all the conditions assayed that were tested using an ANOVA analysis followed by Tukey’s HSD test (*p* < 0.05).

**Figure 3 ijms-26-03481-f003:**
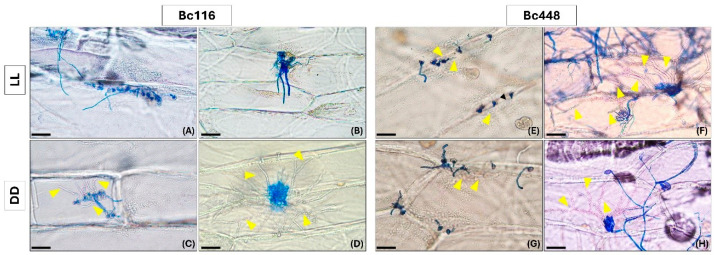
Light affects the ability of isolate Bc116 to penetrate epidermal onion cells. Onion epidermal strips were inoculated with spores (panels (**A**,**C**,**E**,**G**) or with mycelium plugs (panels (**B**,**D**,**F**,**H**) of isolates Bc116 and Bc448 and incubated under LL or DD conditions. Penetration was evaluated visually at 12 hpi for the inoculations performed with spore suspensions and at 24 hpi for the inoculations performed with mycelium plugs. Samples were stained with lactophenol blue. Yellow arrowheads indicate hyphae penetrating host cells. Scale bars, 50 µm in panels (**A**,**C**,**E**,**G**); 100 µm in panels (**B**,**D**,**F**,**H**).

**Figure 4 ijms-26-03481-f004:**
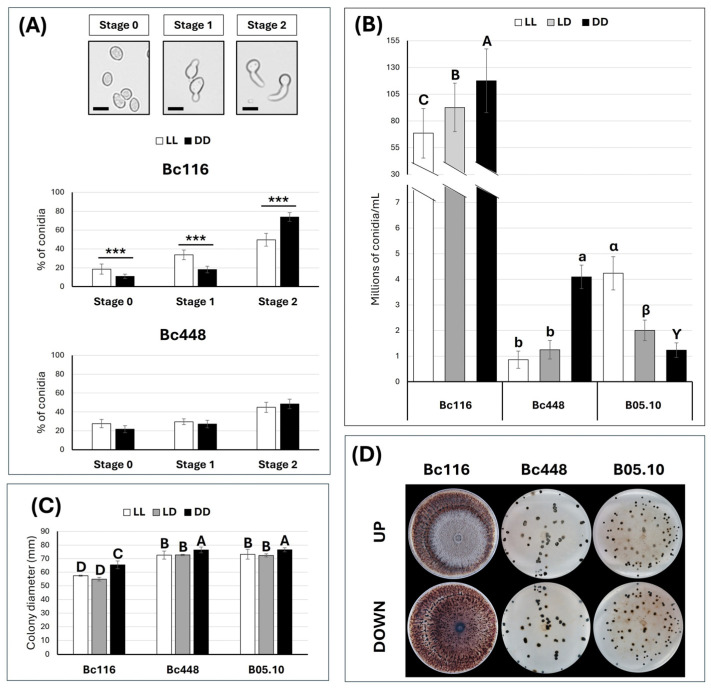
Effect of light on the physiology of isolates Bc116 and Bc448. (**A**) Germination of conidia in liquid medium in static culture. A 60 µL drop of a 5 × 10^5^ sp/mL conidial suspension was prepared and placed in the center of empty Petri dishes. The plates were incubated at 22 °C under different light conditions (LL, white bars; DD, black bars). The number of conidia in each stage (stages 0, 1, 2) was determined 6 hpi. The bars show the media ± SD of three independent biological experiments. For each conidia developmental stage, the percentages of conidia for a pair of measurements (LL and DD) in each strain were tested using a *t*-test and significant differences are indicated by *** (*p* < 0.001). Scale bars in images of spores, 10 µm. (**B**) Effect of light on the conidiation rate of isolates Bc116, Bc448, and B05.10. Mycelium plugs of each fungal isolate were placed in the center of MEA plates and incubated for three weeks at 22 °C under different light conditions (LL, white bars; 16/8 LD, gray bars; DD, black bars). Afterwards, conidia were harvested, and the concentration was estimated using a Thoma cell counting chamber. The bars show the media ± SD of three independent biological experiments. The letters over each bar represent significant differences between conditions assayed for each fungal isolate (uppercase, Bc116; lowercase, Bc448; Greek letters, B05.10) that were tested in independent studies using an ANOVA analysis followed by Tukey’s HSD test (*p* < 0.05). (**C**) Saprophytic growth on synthetic media of isolates Bc116, Bc448 and B05.10. A 10 µL drop of a 5 × 10^5^ sp/mL conidial suspension of each fungal isolate was placed in the center of MEA plates and the diameter of the colony was estimated 5 days post-inoculation (dpi). Plates were incubated at 22 °C and different light conditions (LL, white bars; 16/8 h LD, gray bars; DD, black bars). Bars show the media ± SD of three independent biological experiments. The letters over each bar represent significant differences between all the conditions assayed that were tested using an ANOVA analysis followed by Tukey’s HSD test (*p* < 0.05). (**D**) Production of sclerotia by isolates Bc116, Bc448, and B05.10. Mycelium plugs were placed in the center of MEA plates of 90 mm in diameter that were incubated at 2–4 °C and DD for 4 weeks.

**Figure 5 ijms-26-03481-f005:**
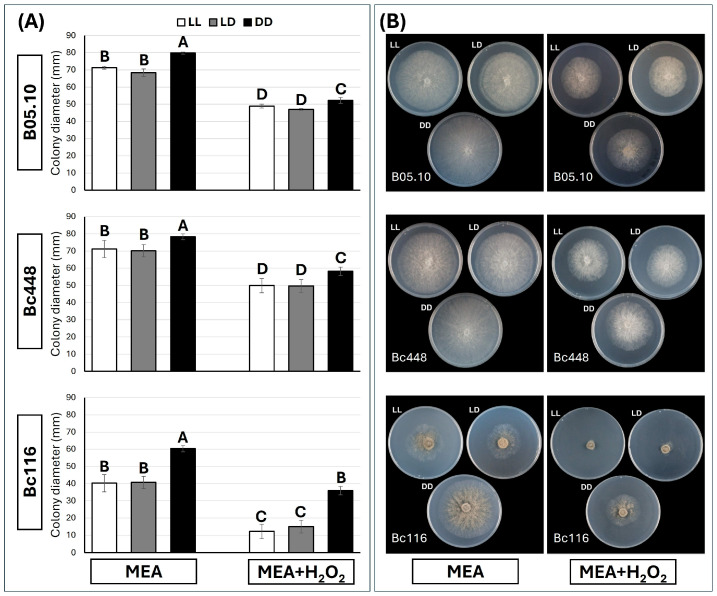
Sensitivity to oxidative stress of isolates Bc116, Bc448, and B05.10. (**A**) Estimation of fungal colony diameter under the conditions assayed. Mycelium plugs from the edge of actively growing colonies were placed in the center of MEA plates of 90 mm in diameter. Oxidative stress was provided by the addition of 7.5 mM H_2_O_2_ to the media. Plates were incubated at 22 °C under different light conditions (LL, white bars; 16/8 h LD, gray bars; DD, black bars) and the colony diameter was estimated 96 hpi. Bars show the media ± SD of three independent biological experiments. The letters over each bar represent significant differences between the conditions assayed for each fungal isolate that were tested in independent studies using an ANOVA analysis followed by Tukey’s HSD test (*p* < 0.05). (**B**) Representative images of colony morphology and colony diameter for each strain and condition are shown.

**Figure 6 ijms-26-03481-f006:**
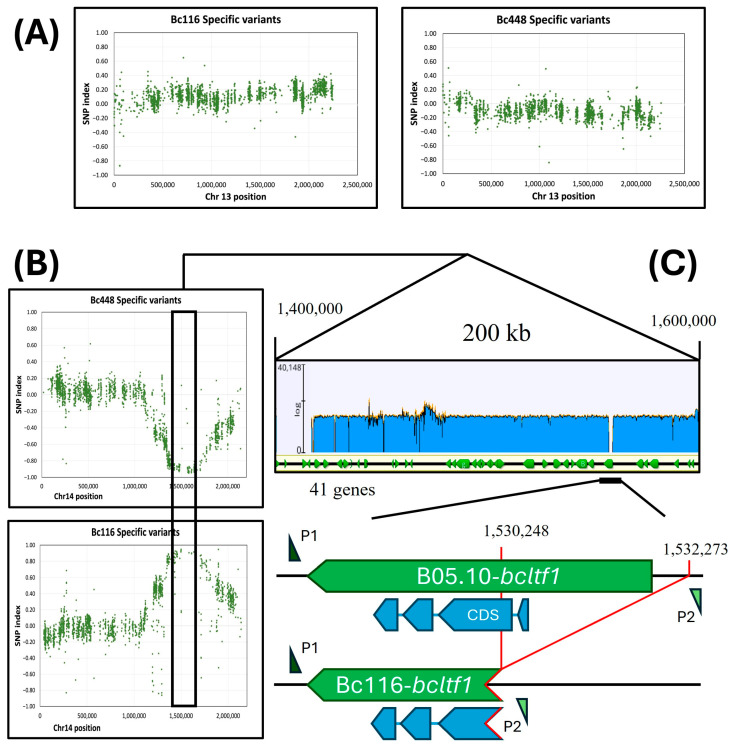
Mapping the genome region linked to the Bc116 phenotype by BSA in the cross Bc116 × Bc448. (**A**,**B**) The figure plots the SNP index of Bc448 specific variants or of Bc116 specific variants (*Y*-axis) across Chr13 (**A**) or Chr14 (**B**) coordinates (*X*-axis). The box in panel (**B**) delimitates a region in Chr14 where the SNP index reaches maximal values for Bc116 specific variants and minimal values for Bc448 specific variants. (**C**) Detailed analysis of the mapped region indicating the genes within this region annotated in the B05.10 genome. The blue graph over the linear representation of the mapped area represents sequencing reads coverage when Bc116 genome reads are aligned with the B05.10 genome. A region with no coverage is observed around the 5′-upstream region of gene *bcltf1*. The annealing positions of the oligonucleotides P1 (*bcltf1-c2F*) and P2 (*bcltf1-c2R*), used to amplify both the B05.10 and Bc116 alleles, are indicated.

**Figure 7 ijms-26-03481-f007:**
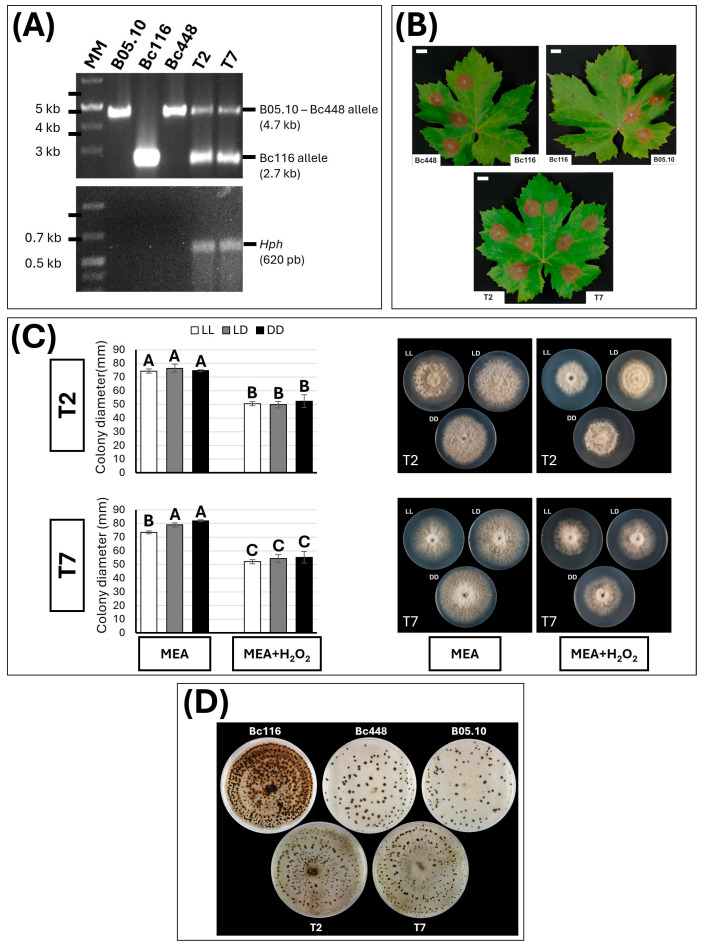
Genetic and phenotypic analysis of Bc116-*bcltf1*-complemented transformants. (**A**) PCR-based analysis of transformants obtained with the B05.10 *bcltf1* allele. Reactions were carried out with genomic DNA of B05.10, Bc116, Bc448, or transformants T2 and T7 as the template. The sizes of the diagnostic bands are indicated. MM: GeneRuler 1 kb Plus DNA Ladder (ThermoFisher Scientific Inc, Waltham, MA, USA). (**B**) Inoculations on *Vitis* leaves of Bc116, B05.10, and Bc448 isolates and transformants T2 and T7. Leaves were inoculated with mycelium plugs of each isolate and incubated at 22 °C under a 16/8 h photoperiod (LD). Images were taken 96 hpi. Scale bars, 20 mm. (**C**) Phenotype of transformants T2 and T7 during growth on MEA plates (4 days at 22 °C under LD conditions) and evaluation of their sensitivity to oxidative stress. Mycelium plugs from the edge of actively growing colonies were placed in the center of 90 mm diameter MEA plates or in MEA plates supplemented with 7.5 mM H_2_O_2_. Plates were incubated at 22 °C under different light conditions (LL, white bars; 16/8 h LD, gray bars; DD, black bars) and the colony diameter was estimated 96 hpi. Bars on the histograms show the media ± SD of three independent biological experiments. The letters over each bar represent significant differences between the conditions assayed for each transformant that was tested in independent studies using an ANOVA analysis followed by Tukey’s HSD test (*p* < 0.05). On the right part of this panel, representative images of colony morphology and colony diameter for each transformant and condition are shown. (**D**) Pattern of production of sclerotia of the fungal isolates on 90 mm diameter MEA plates. Plates were incubated at 2–4 °C under DD for 4 weeks.

**Table 1 ijms-26-03481-t001:** Total number of SNPs identified in the genomes of isolates Bc448 and Bc116 in comparison with the B05.10 genome. The number of SNPs exclusive to each isolate is presented.

	B05.10 Genome	Bc448 SNPs		Bc116 SNPs	
	Chr Size (kb)	Total Number	Exclusive	Total Number	Exclusive
**Chr 1**	4109	23,260	9129	26,619	12,128
**Chr 2**	3341	17,103	5335	19,466	7698
**Chr 3**	3227	13,364	4379	15,161	6176
**Chr 4**	2472	16,024	7069	18,731	9776
**Chr 5**	2959	11,864	4260	12,701	5097
**Chr 6**	2726	17,544	5529	19,339	7324
**Chr 7**	2652	17,831	5715	18,805	6689
**Chr 8**	2617	14,368	5583	15,433	6649
**Chr 9**	2548	16,793	5866	18,543	7616
**Chr 10**	2419	13,662	4484	15,798	6620
**Chr 11**	2360	12,485	3928	15,300	6743
**Chr 12**	2353	13,996	5298	14,802	6104
**Chr 13**	2258	10,921	3949	13,106	6134
**Chr 14**	2138	14,089	5718	15,952	7581
**Chr 15**	2028	11,469	3246	14,335	6112
**Chr 16**	1970	10,795	3556	13,634	6395
**Chr 17**	247	107	74	54	51
**SUM**	**42,424**	**235,675**	**83,118**	**267,779**	**114,893**

## Data Availability

The datasets presented in this study are publicly available in online repositories (SRA of NCBI) under the BioProject PRJNA1224713. (https://www.ncbi.nlm.nih.gov/sra/PRJNA1224713, accessed on 21 March 2025). Accession numbers are as follows: Bc116 Genome—SAMN46863733; Pool aggressive progeny (A)—SAMN46863735; and Pool non-aggressive progeny (B)—SAMN46863734.

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
