# Peer review of "Genetic and Genomic Analysis Identifies bcltf1 as the Transcription Factor Coding Gene Mutated in Field Isolate Bc116, Deficient in Light Responses, Differentiation and Pathogenicity in Botrytis cinerea"

_ijms, 2025, doi:10.3390/ijms26083481_

Round 1

Reviewer 1 Report

Comments and Suggestions for Authors

In this manuscript, Virginia Casado del Castillo and colleagues identified the Botrytis cinerea isolate Bc116 displaying reduced pathogenicity conditioned by the light regime and cloned the BCLTF1 gene encoding the B. cinerea Light responsive Transcription Factor 1. I have following comments:

1, For the title, I suggest to employ “Genetic and genomic analysis identifies bcltf1 as the transcription factor coding gene mutated in field isolate Bc116, deficient in light responses, differentiation and pathogenicity in Botrytis cinerea”.

2, For the Abstract, detailed value in the statement like “This field isolate shows a reduced pathogenicity that is conditioned by the light regime. Light also delays germination and accentuates the negative effect it exerts on the vegetative growth of B. cinerea. Bc116 also displays a marked hyperconidiation phenotype and a characteristic sclerotia production pattern.” should be provided.

3, For the key words, full name of “bcltf1” should be included.

4, For the introduction, required citation should be provided. For instance, please provide the citation for the text from Line 43 to Line 53.

5, For the results, scale bars should be included in Figures 1, 2, 4, 5, and 7.

6, For the materials and methods, genotypes of Botrytis cinerea and bean employed in this study should be described. Biological and technical replicates, as well as randomization methods should be clearly stated. Plant growth conditions and sampling size should be included.

7, For the discussion, I would like to see the discussion section was divided into subsections with appropriate titles.

8, An independent conclusion section should be included.

Author Response

Dear Reviewer

Thank you very much for your comments and suggestions. We find they contribute to improve the clarity of our manuscript. When possible, we have incorporated them or adapted the text.

In this manuscript, Virginia Casado del Castillo and colleagues identified the Botrytis cinerea isolate Bc116 displaying reduced pathogenicity conditioned by the light regime and cloned the BCLTF1 gene encoding the B. cinerea Light responsive Transcription Factor 1. I have following comments:

1, For the title, I suggest to employ “Genetic and genomic analysis identifies bcltf1 as the transcription factor coding gene mutated in field isolate Bc116, deficient in light responses, differentiation and pathogenicity in Botrytis cinerea”.

The title has been adapted taking into consideration the suggestions of reviewer1 and 2. We find that the most appropriate title is:

Genetic and genomic analysis identifies bcltf1 as the Transcription Factor coding gene mutated in field isolate Bc116, deficient in light responses, differentiation and pathogenicity in Botrytis cinerea.

2, For the Abstract, detailed value in the statement like “This field isolate shows a reduced pathogenicity that is conditioned by the light regime. Light also delays germination and accentuates the negative effect it exerts on the vegetative growth of B. cinerea. Bc116 also displays a marked hyperconidiation phenotype and a characteristic sclerotia production pattern.” should be provided.

We apologize, but I am afraid we do not understand exactly the meaning of this suggestion (detailed value in the statement like…. should be provided). The formulation of these statements in the abstract is certainly a short description of the phenotypes displayed by isolate Bc116, which is later considered in detail. We are not sure if more details and values can be presented in the abstract, given the length recommended by the editors. We tried to be precise and concise, and we attempted to present the main information regarding the aspects under consideration.     

3, For the key words, full name of “bcltf1” should be included.

The full name of “bcltf1” has been included in the list of keywords (Botrytis cinerea Light Responsive Transcription Factor 1)

4, For the introduction, required citation should be provided. For instance, please provide the citation for the text from Line 43 to Line 53.

Citations have been included for the text indicated by the reviewer. Some additional references have been added. Following the suggestions offered by the reviewers, the introduction (and the discussion) has been condensed and rewritten in several paragraphs.

5, For the results, scale bars should be included in Figures 1, 2, 4, 5, and 7.

Scale bars have been included in Figures 1, 2, 4, and 7. In those figures whose panels include pictures of plates (Fig. 4, 5, 7, S1), the size of the plates has been indicated in the corresponding figure legend.

6, For the materials and methods, genotypes of Botrytis cinerea and bean employed in this study should be described. Biological and technical replicates, as well as randomization methods should be clearly stated. Plant growth conditions and sampling size should be included.

The information regarding the Botrytis isolates and the bean variety employed in this work is presented in section 4.1 of Materials and Methods. The information about the the B. cinerea B05.10 isolate is presented in the reference by Büttner et al. (1994), and the information regarding the Bc448 and Bc116 isolates is offered in the reference by Acosta Morel et al. (2019). The three isolates are wild type isolates. Only the origin of these isolates can be offered at the moment. Once the nature of the mutation in Bc116 has been determined, it will be possible to offer a reference describing its genotype.  

The bean cultivar Blanca Riñón of Phaseolus vulgaris is a local landrace widely cultivated in Spain. The supplier is also indicated section 4.1 of Materials and Methods. Conditions to grow the plants in the greenhouse (substrate, temperature and light regime) are offered in this section. The sampling size as well as randomization methods in the inoculation experiments are described in section 4.4.

7, For the discussion, I would like to see the discussion section was divided into subsections with appropriate titles.

We have condensed the discussion section and divided it in subsections. This has implied a rewriting of several paragraphs and the reorganization of the entire section.

8, An independent conclusion section should be included.

An independent conclusion section has been included.

Reviewer 2 Report

Comments and Suggestions for Authors

This manuscript presents the discovery of a naturally occurring variant strain of B. cinerea, Bc116, whose pathogenicity is regulated by light. Through genome-based analysis, a gene named bcltf1 was identified, and its mutation in Bc116 was found to result in reduced pathogenicity. The research content is substantial, and the English writing is of good quality. However, there are still some areas that require improvement.

Major comments:

  1. 10 serves as the reference strain, so why was Bc448 used as the control for experiments related to pathogenicity and conidial germination of Bc116, rather than B05.10? Although the authors refer to Bc488 as an "aggressive field isolate" in the manuscript, B05.10 was subsequently introduced as a control in experiments concerning conidial production capacity, hyphal growth, sclerotium formation, and antioxidant properties. This inconsistency in controls may lead to misunderstandings among readers.
  2. Figure 4D, Why were the LL, LD, and DD treatments not conducted during the sclerotium formation assay, whereas these three conditions were set for both colony growth and conidial production?
  3. The layout of Figure 5 is difficult to understand; it is recommended to separate the colony images and statistical graphs into panels A, B, C, and D, etc. for better organization.
  4. Lanes 342-344: what dose tiny sclerotia mean? Dose it means this kind of sclerotia cannot produce ascospores after fertilized by microconidia? Why established using Bc116 as the spermatizing strain? In addition, the title of this part (2.5) is not appropriate.
  5. Lanes 396-411: the description of variation analysis in Bc116 genome is better to move to the previous section 2.6.
  6. In 2.7, Some phenotypes like conidiation, germination of conidia, and sensitive to oxidative stress that determinated previously are missing in T2 and T7.
  7. The presence of B05.10-Δbcltf1 mutants significantly diminishes the novelty of this manuscript. Given that Bc116 exhibits a phenotype remarkably similar to that of B05.10-Δbcltf1 mutants, upon the discovery of the Bc116 strain, one could have directly employed PCR and RT-PCR methods to detect mutations and expression levels of the bcltf1 gene. There was no necessity to invest considerable time in genome-based SNP detection. Therefore, it is recommended that the authors, when presenting their results, focus on the differential phenotypes between Bc116 and B05.10-Δbcltf1 mutants, include analyses of other significant SNPs, measure the expression levels of related genes, and engage in relevant discussions.

Minor comments:

  1. The title of this manuscript is suggested to change as ‘Genomic analysis revealed the natural mutation of bcltf1 resulting deficient in light responses, vegetative growth, conidiation, and pathogenicity in field isolate Bc116 of Botrytis cinerea
  2. Some sections of the introduction and discussion are overly verbose and could be appropriately condensed, such as lines 54-102, lines 443-458, and lines 494-548.

Author Response

Dear reviewer:

Thank you very much for your comments and suggestions. We find they contribute to improve the clarity of our manuscript. When possible, we have incorporated them or adapted the text.

Reviewer 2

Comments and Suggestions for Authors

This manuscript presents the discovery of a naturally occurring variant strain of B. cinerea, Bc116, whose pathogenicity is regulated by light. Through genome-based analysis, a gene named bcltf1 was identified, and its mutation in Bc116 was found to result in reduced pathogenicity. The research content is substantial, and the English writing is of good quality. However, there are still some areas that require improvement.

Major comments:

  1. 10 serves as the reference strain, so why was Bc448 used as the control for experiments related to pathogenicity and conidial germination of Bc116, rather than B05.10? Although the authors refer to Bc488 as an "aggressive field isolate" in the manuscript, B05.10 was subsequently introduced as a control in experiments concerning conidial production capacity, hyphal growth, sclerotium formation, and antioxidant properties. This inconsistency in controls may lead to misunderstandings among readers.

We think that the information that natural field isolates can offer in the analysis of the fungal pathogen and of the pathosystems in which it participates can be very interesting and extends that derived from the analysis of the reference strains. It is for this reason that we work with the B. cinerea natural populations from our vineyards. Bc448 was initially selected because it belongs to the same “natural population” as Bc116 (as indicated in the text), it has been shown previously to resemble the reference isolates in aspects regarding to pathogenicity, morphotype and growth characteristics (as indicated in section 2.5), and it was going to be used in the genetic analysis as the male parent isolate (in previous works -Acosta Morel et al., 2021- this isolate was successfully used in crosses). Bc116 and Bc448 have, for these reasons, been included in the physiological analysis. Then, to gain information comparing the field isolates with a reference isolate, we decided to include the B05.10 isolate. As indicated in section 2.3, it has been found that Bc448 and B05.10 display similar growth characteristics and light responses, but specifically the stimulatory effect of light in sporulation observed in B05.10 is not found Bc448, which in this sense resembles more the Bc116 isolate.

To clarify the reason for the inclusion of the B05.10 isolate in those analyses, the text has been rewritten as follows: “To investigate it, we determined the production of spores by isolate Bc116 in MEA plates in comparison with the field isolate Bc448 under the three light regimes, LL, LD and DD. It was decided to include in this analysis the B05.10 strain to gain information in the comparison with a reference isolate” (lines 271-273).

  1. Figure 4D, Why were the LL, LD, and DD treatments not conducted during the sclerotium formation assay, whereas these three conditions were set for both colony growth and conidial production?.

The reason is that those, DD and low temperatures, are the conditions described in literature as the conditions stimulating the production of sclerotia, and these are the conditions we routinely use to produce sclerotia in the lab to make crosses. The evaluation of the ability of the three isolates under different light regimes could be done in the future to extend their characterization. But it is true that in this work we have just considered the described optimal conditions for sclerotia production. What we observed, and we consider that it pinpoints and important difference with the phenotype of the B05.10-Δbcltf1 mutants, is that Bc116 produces sclerotia, with a morphology and a pattern of distribution different from that observed in the B05.10 and the Bc448 isolates. And this is the main point we raise in this context in the Results and Discussion sections.

  1. The layout of Figure 5 is difficult to understand; it is recommended to separate the colony images and statistical graphs into panels A, B, C, and D, etc. for better organization.

The layout of Figure 5 has been modified. A panel A has been prepared with the histograms, and a panel B has been prepared with the images. We hope this organization helps to visualize and understand the analysis performed. Figure legend has been adapted accordingly.

  1. Lanes 342-344: what dose tiny sclerotia mean? Dose it means this kind of sclerotia cannot produce ascospores after fertilized by microconidia? Why established using Bc116 as the spermatizing strain? In addition, the title of this part (2.5) is not appropriate.

Yes. The tiny, very small sclerotia can’t be handled and used in crosses. The routine to make crosses, in our hands, and as recommended by other authors (Rodenburg et al., (2018). Functional analysis of mating type genes and transcriptome analysis during fruiting body development of Botrytis cinerea. MBio, 9(1), 10-1128), is to use large sclerotia (at least 5 mm in diameter, better if larger) for them to support the development of apothecia, if fertilization is effective. This limitation is what we point out in the text in lines 336-337. 

Regarding the title of section 2.5, we could consider better options but, sincerely, we think the title reflects the information being presented in the section. A cross between the isolates of contrasting phenotypes is established, the progeny is phenotyped and the segregation of the traits under investigation is studied. We find this is strictly a genetic analysis (sensu strictu) and for that reason we use this title: “Genetic analysis of Bc116”. Perhaps it is more informative indicating “Genetic analysis of the Bc116 phenotype”. We have included this formulation.  

  1. Lanes 396-411: the description of variation analysis in Bc116 genome is better to move to the previous section 2.6.

The information has been moved to section 2.6

  1. In 2.7, Some phenotypes like conidiation, germination of conidia, and sensitive to oxidative stress that determinated previously are missing in T2 and T7.

We used the phenotypes described in the text, the recovery of aggressiveness on Vitis leaves in LD, mycelial growth rate, the loss of the premature sporulation capacity and the pattern of sclerotia production of Bc448 and B05.10, as informative of functional complementation, as they are all phenotypes under the control of BcLTF1, as those regarding the other traits, and these are most easily scorable. We did not collect data to quantify conidiation, but before submission we had made a partial characterization of the sensitivity of T2 and T7 to oxidative stress. We have extended these analyses during these 10 days (the revision) including additional replicas to evaluate precisely their sensitivity to oxidative stress and we have collected the data we are presenting now in the form of a new version of panel C. In this panel we show the aspect of the T2 and T7 colonies when cultured on MEA plates, for comparison regarding the radial growth and the early sporulation phenotype with the colonies of Bc116, Bc448 and B05.10 presented in Fig. 5B, and the aspect of the T2 and T7 colonies when cultured in MEA plates in the presence of H2O2, also to be compared with the colonies of the Bc116, Bc448 and B05.10 presented in Fig. 5B, now regarding the sensitivity to oxidative stress. The text in section 2.7 and the figure legend has been adapted accordingly. The analysis presented now is wider and more informative.  

  1. The presence of B05.10-Δbcltf1 mutants significantly diminishes the novelty of this manuscript. Given that Bc116 exhibits a phenotype remarkably similar to that of B05.10-Δbcltf1 mutants, upon the discovery of the Bc116 strain, one could have directly employed PCR and RT-PCR methods to detect mutations and expression levels of the bcltf1 gene. There was no necessity to invest considerable time in genome-based SNP detection. Therefore, it is recommended that the authors, when presenting their results, focus on the differential phenotypes between Bc116 and B05.10-Δbcltf1 mutants, include analyses of other significant SNPs, measure the expression levels of related genes, and engage in relevant discussions.

We understand the point raised by the reviewer. Let us mention that once the phenotype of Bc116 had been preliminary characterized, and once its genetic analysis had been initiated (crosses established), we considered that there were several genes whose alteration could give rise to these kinds of phenotypes. In fact, we had a first candidate, the bcwcl1 gene, that we were aware that displayed similar phenotypes to Bc116. But once recovered and sequenced, we did not find any alteration in the Bc116 bcwcl1 allele that could explain the alterations. There could be other options and candidates, it is true, but we considered that it would be worthwhile applying a genetic approach without any preconception. It is not the first time that new genetic factors are brought to light by analysing mutants displaying phenotypes similar to other previously reported (we have an experience on this working with mycelial non-aggressive isolates which led to the identification of the B. cinerea ortholog of the N. crassa ff-7 gene -Acosta Morel et al., 2021-). We then considered that a genetic approach would be informative. Furthermore, and this is something we wanted to highlight in this manuscript, which is being submitted to an issue focussing on current advances on genetic and genomic tools in microbial genetics, we find it is feasible nowadays going from the phenotype to the gene within a reasonable period of time and with an affordable economic investment. The genetic and genomic analysis unequivocally demonstrate that Bc116 is altered in bcltf1.

Following the suggestions offered by the reviewers, we have condensed and rewritten the discussion focussing on the differential phenotypes of Bc116 and on the information its analysis is offering about the functions of BcLTF1.

This work is already being extended in the lab focussing on the analysis of genes under the control of BcLTF1 and considering the analysis of double mutants in combinations of bcltf1 with bcwcl1 and with Bcin04g03490, as experimental evidence suggest that the three TFs participate in the regulation of light and pathogenicity pathways in B. cinerea. But all these aspects need further, and specific work already initiated.

Minor comments:

  1. The title of this manuscript is suggested to change as ‘Genomic analysis revealed the natural mutation of bcltf1 resulting deficient in light responses, vegetative growth, conidiation, and pathogenicity in field isolate Bc116 of Botrytis cinerea

Two reviewers suggested modifications in the title. The authors have agreed in the following formulation:

“Genetic and genomic analysis identifies bcltf1 as the Transcription Factor coding gene mutated in field isolate Bc116, deficient in light responses, differentiation and pathogenicity in Botrytis cinerea”

  1. Some sections of the introduction and discussion are overly verbose and could be appropriately condensed, such as lines 54-102, lines 443-458, and lines 494-548.

We have attempted to condense several sections of the introduction and of the discussion. Unifying the suggestions offered by the reviewers we have rewritten several paragraphs in both sections and reorganized the discussion in subsection. We hope these changes ease the transmission of information and clarify it.

Reviewer 3 Report

Comments and Suggestions for Authors

Author Response

Dear reviewer:

Thank you very much for your comments and suggestions. We find they contribute to improve the clarity of our manuscript. When possible, we have incorporated them or adapted the text.

Reviewer 3

 In the paper entitled “Genetic and genomic analysis identifies bcltf1 as the TF coding gene mutated in field isolate Bc116, deficient in light responses, differentiation and pathogenicity in Botrytis cinerea”, the authors characterize the pathogenicity of a field isolate under different light conditions.

Although the role of BcLTF1 in pathogenicity was already described, as the authors know, the paper provides a thorough characterization of real-world isolates affected in bcltf1 further supporting the previous findings.

In any case, I would like the authors to address a few questions before publication.

Major comments:

-The method’s section regarding the bioinformatics is very sparse. It would be impossible to reproduce the results without a better description of the methods. I understand that the authors use a commercial software (Geneious) which to the best of my knowledge is a wrapper for many commonly used bioinformatic tools. For example, which mapper was used? (bowtie2? BWT?), which snp-caller? (Freebayes?), did the authors perform adapter trimming? (trimmomatic).

We have rewritten section 4.9 indicating the tools used. Trimming and quality control was performed with FASTQ by the sequencing company. The clean data were mapped with Bowtie2. The Geneious SNP calling tool was used to generate the SNP lists.

-The legend used to show the statistical differences in the figures is confusing. For example, in Figure 4A lower panel. What does the CD letters mean? I understand that the same letter means a group with no statistical differences among the members of the group. But what is the span of the letters? Do the group A of the Fig4A Bc116 is the same as the group A of the same figure Bc448? Same applies to Fig 4B. Is group A different for each bar shown. It would be easier to interpret the results using lines connecting all interesting comparisons and symbols to show the p-value.

Figure 4 legend has been rewritten including information clarifying the statistics analysis performed in each case.  In panel A, in the new submitted version of our manuscript, a “t-test” has been performed for each developmental stage in each isolate in the two light regimes considered. Significant differences in the comparison of light conditions are indicated with asterisks. In panel B, differences between light conditions for each strain have been considered. Different letter codes have been used for each isolate. In panel C, data from the three isolates in the three light regimes are analysed together. This is specifically indicated in the figure legend for each panel.

-In my opinion, the discussion section is too long. It contains repetitive information already presented in the results section. I suggest the authors to short the discussion section and focus on the most relevant elements.

Following the recommendations of the reviewer, the discussion has been condensed. Unifying the comments by the reviewers, it has been reorganized and divided in subsections. We hope these changes ease the transmission of information and clarify it.

-Do the authors think that mutations in bclft1 are relevant for the fitness of the pathogen? Are they commonly found? How common is it to find isolates with lower pathogenetic? Could this mutation an adaptation to certain conditions? These questions should be addressed in the discussion.

This is an interesting point that, following the suggestions received, we have attempted to address it in the manuscript. A section in the discussion has been added. 

-During the genetic analysis the authors found a 1:1 segregation pattern. Does it mean that all the descendants are identical in terms of aggressiveness? Are there any descendant with differences in the penetrance of the traits? I would expect that a mutation in such a critical gene for the lifestyle of the pathogen would drive the selection and fix of additional compensatory mutations within the genome of the pathogen. If that is the case the descendants of the pathogen would show differences in the penetrance of the pathogenicity.

This is a very nice observation. Regarding the alterations in pathogenicity, we must say that we have quantified the aggressiveness in the aggressive progeny and remarkably, large differences among individuals were observed (which can be observed in the collection of inoculated leaves presented in Fig. S1, panel A). We understand that this observation indicates that the functional wild type allele of bcltf1, which controls light responses, development and sensitivity to oxidative stress, also controls a number of pathogenicity factors. Bc448 and Bc116 are likely genotypically different regarding some, or many, of these factors which segregate in the progeny, determining differences in the level of aggressiveness (in the penetrance of this trait) when a functional bcltf1 allele is present. A similar situation has been described for the gene Bcin04g03490 previously identified by our group, a major effect gene controlling development and pathogenicity in B. cinerea (Acosta Morel et al., 2021). A comment describing this situation has been added in section 2.5. A histogram with data quantifying the aggressiveness of one third of the progeny collected and analysed in this work has been included in Fig. S1, panel B.

Round 2

Reviewer 1 Report

Comments and Suggestions for Authors

Authors have addressed my concerns in the revision.

Reviewer 2 Report

Comments and Suggestions for Authors

Accept in present form

Reviewer 3 Report

Comments and Suggestions for Authors

The authors have addressed all my comments, and I believe the manuscript is ready for publication.